# COMBA: CROSS BATCH AGGREGATION FOR LEARNING LARGE GRAPHS WITH CONTEXT GATING STATE SPACE MODELS

## ABSTRACT

State space models (SSMs) have recently emerged for modeling long-range dependency in sequence data, with much simplified computational costs than modern alternatives, such as transformers. Advancing SMMs to graph structured data, especially for large graphs, is a significant challenge because SSMs are sequence models and the shear graph volumes make it very expensive to convert graphs as sequences for effective learning. In this paper, we propose COMBA to tackle large graph learning using state space models, with two key innovations: graph context gating and cross batch aggregation. Graph context refers to different hops of neighborhood for each node, and graph context gating allows COMBA to use such context to learn best control of neighbor aggregation. For each graph context, COMBA samples nodes as batches, and train a graph neural network (GNN), with information being aggregated cross batches, allowing COMBA to scale to large graphs. Our theoretical study asserts that cross-batch aggregation guarantees lower error than training GNN without aggregation. Experiments on benchmark networks demonstrate significant performance gains compared to baseline approaches. Code and benchmark datasets will be released for public access.

## 1 INTRODUCTION

Graph learning has become popular to handle data structures for representing entities and their relationships in a wide range of domains, including social networks (Fan et al., 2019), citation networks (Yang et al., 2016), and molecular structures (Wu et al., 2018). Graph neural networks (GNNs) have achieved great success in learning from graph-structured data by propagating neighborhood information. However, scaling GNNs to large graphs remains challenging due to the high memory and computational cost of neighborhood expansion and the difficulty of preserving long-range dependencies (Dwivedi et al., 2022).

Transformers, by contrast, excel at modeling long-range dependencies through global attention, which directly relates every node to every other node in the graph (Yun et al., 2019). While global attention enables Transformers to capture arbitrary dependencies, graphs lack the inherent sequential ordering. To address this, graph Transformers incorporate structural encodings (SEs) or positional encodings (PEs) to provide nodes with contextual information about their position and role within the graph (Kim et al., 2022). However, the quadratic cost of attention makes Transformers difficult to apply at scale, particularly for massive graphs with millions of nodes and edges. To mitigate this, some works have proposed linear-time attention approximations, such as sparse attention (Zaheer et al., 2020) and low-rank factorization (Wang et al., 2020), which reduce memory and computational costs compared to full graph attention.

Recently, state space models (SSMs) have emerged as efficient alternatives for sequence modeling, combining structured recurrence with linear-time complexity while still capturing long-range dependencies (Gu et al., 2021). Mamba is a state space model (SSM) architecture designed to efficiently model long-range dependencies in sequential data. Unlike traditional models that rely on attention mechanisms, Mamba employs a selective state space framework, enabling it to process sequences in linear time while maintaining high performance across various modalities (Gu & Dao, 2024).

Yet, applying SSMs to graph data poses unique challenges. First, graphs are inherently irregular and lack a natural sequential ordering, making it difficult to directly feed them into sequence models. Second, traditional SMMs process sequences by updating the current hidden state, which is sufficient for sequential data but inadequate for graphs where each node depends on multi-hop neighborhoods. Since the sheer size of graphs demand scalable and efficient learning, we adopt a batch-based sampling strategy, which introduces additional challenges such as sampling bias and variance (Balaji et al., 2025).

To tackle these challenges, our COMBA framework incorporates three key components: (1) hop-aware graph context construction, which leverages adjacency matrices at multiple hops to capture local and multi-hop neighborhood information; (2) cross-batch aggregation, which updates node embeddings across overlapping batches to mitigate sampling variance and preserve global graph information; and (3) graph context gating, which selectively controls the contribution of each hop to a node's representation. Together, they allow COMBA to efficiently model long-range dependencies in large graphs. By combining local hop-aware context with cross-batch updates, the framework balances scalability with expressive power, enabling robust learning on large homogeneous graphs.

**Specific contributions** of this paper are as follows:

- We develop a cross-batch aggregation mechanism to mitigate variance from batch-based sampling and preserve global information.

- We introduce graph context gating, which selectively controls the contribution of multi-hop neighborhoods to each node's representation.

- We propose COMBA, a state-space model framework for learning from large-scale homogeneous graphs, with validation on benchmark datasets demonstrating significant performance gains over baselines.

- Our theoretical study asserts that cross-batch aggregation reduces error compared to traditional batch-wise GNN training without cross-batch aggregation.

## 2 RELATED WORK

### 2.1 HOMOGENEOUS GRAPH NEURAL NETWORKS

Graph neural networks (GNNs) have emerged as a powerful tool for learning on graph-structured data. Pioneer works such as Graph Convolutional Networks (GCNs) (Kipf & Welling, 2017), extended convolutional operations to graphs by aggregating information from a node's neighbors. Other works, including GAT (Velickovic et al., 2017), incorporate attention mechanisms to assign different weights to different neighbors, and GatedGCN (Bresson & Laurent, 2017) extends standard GCNs by using learnable edge-wise gates to control neighbors' influence during message passing.

### 2.2 GRAPH TRANSFORMERS

Graph Transformers extend the standard Transformer architecture to handle graph-structured data by integrating both node features and graph topology into the attention mechanism. The first generalized graph transformer (Dwivedi & Bresson, 2020) incorporates neighborhood connectivity as their attention mechanism, and introduces Laplacian-based positional encodings. Graphormer (Ying et al., 2021) encodes both node features and graph structure, using spatial encodings and centrality-aware attention to capture relationships between nodes. NAGPhormer (Chen et al., 2022) uses node-adaptive gating and hierarchical attention over multi-hop neighborhoods to better capture long-range dependencies in graphs. Although graph transformers are highly expressive and capable of capturing long-range dependencies, their quadratic complexity with respect to the number of nodes limits scalability to large graphs.

### 2.3 STATE SPACE MODELS WITH GRAPHS

State space models (SSMs) have emerged as a powerful alternative to traditional recurrent architectures for sequence modeling. Structured State Space Models (S4) are a class of state space

models designed to efficiently model long-range dependencies in sequential data using reparameterization (Gu et al., 2021). Recently, Mamba (Gu & Dao, 2024) demonstrated fast inference and linear-time complexity in long sequence modeling with improved performance compared with Transformers. In the context of graphs, Mamba has been adapted to propagate information along nodes and edges, allowing a expressive node representation across multi-hop neighborhoods. A recent example is Graph Mamba (Wang et al., 2024) which integrates a Mamba block with graph-specific node prioritization and permutation strategies to efficiently capture long-range dependencies. Another Graph Mamba (Behrouz & Hashemi, 2024) considers nodes' induced subgraphs as tokens and generates input sequences for Mamba by applying MPNNs to these subgraphs. While effective, its reliance on computing MPNNs over all node subgraphs can be computationally expensive, motivating our batched variant and modifications such as the context gating mechanism for improved efficiency and performance.

## 2.4 HOMOGENEOUS GRAPH SCALING

Scaling GNNs to large graphs remains a challenge, primarily due to the neighborhood explosion problem, where the number of nodes involved in computation grows exponentially with the number of layers. To mitigate this problem, GraphSAGE Hamilton et al. (2017) generates node embeddings by sampling and aggregating features from a fixed-size neighborhood, enabling generalization to unseen nodes while reducing computational cost compared to full-graph methods. Similarly, Clustering-based strategies offer another solution by partitioning the graph into smaller, more manageable subgraphs. Cluster-GNN Chiang et al. (2019) divides the graph using a clustering algorithm (such as Metis) and trains a GNN with mini-batch updates on these clusters, which significantly reduces memory usage. FastGCN Chen et al. (2018) further improves efficiency by using importance sampling to train on large graphs and treats the convolutions as integral transforms, thereby approximating the feature propagation through Monte Carlo sampling.

Our approach utilizes adjacency matrices from different hops to construct hop-aware graph context for each batch. For each hop, we apply a graph neural network (GNN) to the corresponding adjacency matrix, enabling the model to capture structural information from multiple neighborhood ranges. To prevent isolated batch training from losing global information, we propose a cross-batch aggregation mechanism to help all batches to update their node embeddings, through shared multi-hop neighbors between batches. To adaptively balance the contributions of hop-aware graph contexts, we use a learnable gating function that selectively weights hop embeddings, allowing the model to emphasize more informative hops. Together, adjacency-driven hop contexts, cross-batch aggregation, and gating for adaptive control enable our model to effectively scale state space models to large graphs, while preserving both local and long-range dependencies.

## 3 PROBLEM DEFINITION

Let $G = (V, E, X)$ be a homogeneous graph, where the node set is $V = \{v_1, v_2, \ldots, v_n\}$, the edge set is $E \subseteq V \times V$, and $X \in \mathbb{R}^{n \times d}$ is the node feature matrix, with each row $x_i$ denoting the $d$-dimensional feature of node $i \in V$. We denote $A \in \{0, 1\}^{n \times n}$ as the binary adjacency matrix of $G$, defined as

$$A_{ij} = \begin{cases} 1 & \text{if } (v_i, v_j) \in E, \\ 0 & \text{otherwise,} \end{cases} \quad A_{ii} = 0 \quad \text{for all } i.$$

Below we also define the hop-$k$ adjacency matrices which indicate whether two nodes are connected by a path of exactly length $k$.

$$\text{Gen}(G, A) = A^k, \quad \text{where}$$

$$A_{ij}^k = \begin{cases} 1, & \text{if } (A^k)_{ij} > 0 \text{ and } (A^r)_{ij} = 0 \ \forall r = 1, \ldots, k-1, \\ 0, & \text{otherwise,} \end{cases} \quad (1)$$

$$A_{ii}^k = 0, \quad \forall i.$$

**Structured State Space Model (S4)** The Structured State Space (S4) model is a sequence model designed to capture long-range dependencies efficiently. For a sequence of inputs $x_t \in \mathbb{R}^N$, we can

map the sequence to output sequence $y(t) \in \mathbb{R}^N$ by the latent state $h(t) \in \mathbb{R}^N$. The continuous-time linear state space formulation is:

$$h'(t) = \mathbf{A}h(t) + \mathbf{B}x(t), \tag{2}$$
$$y(t) = \mathbf{C}h(t), \tag{3}$$

where $\mathbf{A} \in \mathbb{R}^{N \times N}, \mathbf{B} \in \mathbb{R}^N, \mathbf{C} \in \mathbb{R}^N$ are the system matrices representing the state transition, input projection, and output projection, respectively. Since the real-world data is usually discrete, we can discretize the continuous system using the zero-order hold (ZOH) discretization rule, yielding:

$$h_{t+1} = \bar{\mathbf{A}}h_t + \bar{\mathbf{B}}x_t, \tag{4}$$
$$y_t = \mathbf{C}h_t, \tag{5}$$

where $\bar{\mathbf{A}} = \exp(\Delta\mathbf{A}), \bar{\mathbf{B}} = (\Delta\mathbf{A})^{-1}(\exp(\Delta\mathbf{A} - I)\Delta\mathbf{B}$, are the discretized matrices, and $\Delta$ is the discretization step size.

S4 has demonstrated strong performance on long sequences due to its ability to model global dependencies, while remaining computationally efficient (Gu et al., 2021). Our **goal** is to design an effective state space model-based learning framework for large graphs with maximum node classification accuracy.

## 4 PROPOSED FRAMEWORK

Our model is primarily motivated by Graph Mamba (Behrouz & Hashemi, 2024), which extends state space models (SSMs) to the graph domain by converting graph neighborhoods into sequential structures. Each node's representation is updated by applying an message-passing neural network (MPNN) to subgraphs generated from random walks starting from that node, followed by Mamba-style sequence modeling to capture long-range dependencies. This design introduces sequential inductive bias into graph representation learning, allowing the model to better combine local structural information and sequential dependencies. However, Graph Mamba also suffers several limitations: (1) it performs MPNN computations on every node's random-walk subgraphs, which becomes computational expensive on large-scale graphs, and (2) it uses only the current input to compute the gating signal, potentially missing richer contextual dependencies from adjacent graph contexts.

In contrast, our framework employs a batch-based hop-aware adjacency construction, where each batch directly encodes multi-hop neighborhoods through MPNN, avoiding redundant per-node subgraph processing. Specifically, COMBA integrates two core components to overcome the limitations of prior approaches: (1) a cross-batch aggregation promotes interactions between batches, (2) a graph context gating to selectively control multi-hop information flow.

### 4.1 LOCAL GRAPH CONTEXT

Let $\mathcal{B} = \{b_1, b_2, \ldots, b_{\hat{m}}\}$ denote a set of batches of nodes. For each batch $b_m \in \mathcal{B}$ we extract the induced subgraph $\mathcal{G}_{b_m}$ to enable localized and scalable computation. Based on $\mathcal{G}_{b_m}$, we then generate the sequence of hop-based adjacency matrices:

$$A_{b_m} = \{A_{b_m}^1, A_{b_m}^2, \ldots, A_{b_m}^{\hat{k}}\} = \{\text{Gen}(\mathcal{G}_{b_m}, A_{b_m}), \text{Gen}(\mathcal{G}_{b_m}, A_{b_m}^2), \ldots, \text{Gen}(\mathcal{G}_{b_m}, A_{b_m}^{\hat{k}})\},$$

where $\mathcal{G}_{b_m}$ and $A_{b_m}$ are the subgraph and adjacency matrix restricted to nodes in batch $b_m$, and $\text{Gen}(\cdot)$ is the hop-matrix generation function defined in Eq. 1. These matrices capture multi-hop connectivity patterns within each batch, and $\mathcal{A} = \{A_{b_1}, \ldots, A_{b_{\hat{m}}}\}$ contains sequences of the hop matrices for all batches.

For each each batch $b_m$, a graph learner is trained, using the hop-based adjacency matrices $\{A_{b_m}^k\}_{k=1}^{\hat{k}}$ together with the corresponding node features $X_{b_m}$. This produces hop-aware node representations for each batch:

$$\mathcal{Z}_{b_m} = \{Z_{b_m,1}, Z_{b_m,2}, \ldots, Z_{b_m,\hat{k}}\}, \quad Z_{b_m,k} = \sigma\left(A_{b_m}^k X_{b_m} W_{b_m}\right)$$

allowing the models to capture structural information at different hop distances within the batch.

Concatenating over all batches yields the global representation set $\mathcal{Z} = \{\mathcal{Z}_{b_1}, \mathcal{Z}_{b_2}, \ldots, \mathcal{Z}_{b_{\hat{m}}}\}$.

To capture local structural context, we utilize the hop dimension in $\mathcal{Z}$. For each node $n$, its sequence of hop-aware embeddings is $\left\{ \mathcal{Z}_{n,1}, \ \mathcal{Z}_{n,2}, \ \ldots, \ \mathcal{Z}_{n,\hat{k}} \right\}$, which encodes information aggregated from neighborhoods of different ranges.

We then define a local hop-context window around hop $k$ with window size $w$ as $\mathcal{C}_n^k = \left\{ \mathcal{Z}_{n,\, k-w}, \ldots, \mathcal{Z}_{n,\, k+w} \right\}$, ensuring that each node representation at hop $k$ incorporates information from both its own embedding and its neighboring hops.

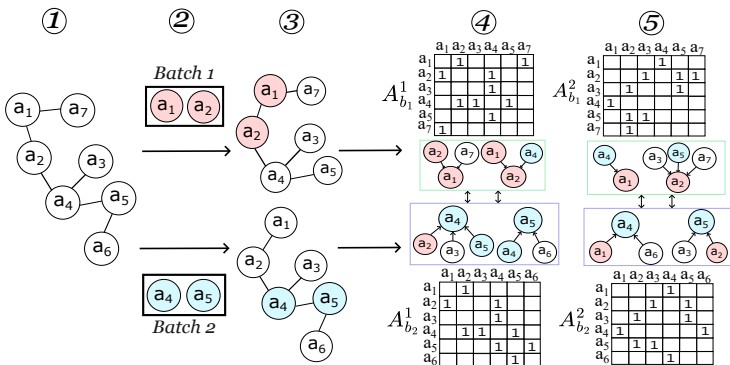

Figure 1: Cross batch aggregation process. From left to right, given a graph in ①, the nodes are partitioned into batches (only two batches are shown in ②). For each batch, COMBA first finds each nodes' $\hat{k}$-hop neighbors and forms a subgraph shown in ③ ($\hat{k} = 2$ in this case). A GNN is trained for each 1-hope, 2-hop, and $k$-hop based adjacency matrix. When training each hop's GNN, information from other batches are used to help learn current batch node's embedding. *E.g.*, in ④, node $a_2$ in Batch 1 aggregates information from $a_4$ from Batch 2. Cross batch aggregation allows all GNNs being trained to collectively help each others.

## 4.2 CROSS BATCH AGGREGATION

To compute node representations for each batch, we first apply Dropout to the input feature matrix for all nodes in the batch and perform a linear transformation:

$$\tilde{X}_{b_m} \ = \text{Dropout}(X_{b_m}, p) \tag{6}$$

$$Z_{b_m}^{(0)} = \sigma(\tilde{X}_{b_m} W_{b_m}^{(0)}) \tag{7}$$

where, $\text{Dropout}(X_{b_m}, p)$ randomly zeroes elements of $X_{b_m}$ with probability $p$. $W_{b_m}^{(0)}$ is a learnable weight matrix, and $\sigma(\cdot)$ is a nonlinear activation function.

For nodes in batch $b_m$, the message passing at layer 1 and hop $k$ is computed as

$$Z_{b_m,k}^{(1)} = \sigma\big(A_{b_m}^k Z_{b_m}^{(0)} W_{b_m}^{(1)}\big), \tag{8}$$

where $A_{b_m}^k$ is the hop-based adjacency matrix. $W_{b_m}^{(1)}$ is the learnable weight for batch $b_m$ with hop $k$ at layer 1 and $\sigma(\cdot)$ is a non-linear activation function such as ReLU. The resulting $Z_{b_m,k}^{(1)}$ contains the embeddings for all nodes in batch $b_m$ under hop $k$. The updated hop-$k$ representations of each node $n$ in batch $b_m$ across all batches that contain that node are then explicitly updated as

$$Z_{b_{1:\hat{m}},k}^{(0)}(n) \ \leftarrow \ Z_{b_m,k}^{(1)}(n), \quad \forall n \in b_m \cap b_{1:\hat{m}}, \ \ k = 1, \ldots, \hat{k} \tag{9}$$

For deeper layers $(l+1)$, the embedding is computed recursively by the cross-batch updated node embeddings $Z_{b_m,k}^{(0)}$:

$$Z_{b_m,k}^{(l+1)} = \sigma\big(A_{b_m}^k Z_{b_m,k}^{(0)} W_{b_m}^{(l+1)}\big), \tag{10}$$

$$Z_{b_{1:M},k}^{(0)}(n) \ \leftarrow \ Z_{b_m,k}^{(l+1)}(n), \quad \forall n \in b_m \cap b_{1:\hat{m}} \tag{11}$$

Fig. 1 shows an example of information aggregation across two batches, where node $a_2$ from Batch 1 leverages information from $a_4$, which is in Batch 2, to learn embeddings.

Finally, for each batch $b_m$, we obtain a sequence of $\hat{k}$ embeddings corresponding to the $\hat{k}$ hop-based adjacency matrices:

$$Z_{b_m} = \{Z^{(L)}_{b_m,1}, Z^{(L)}_{b_m,2}, \ldots, Z^{(L)}_{b_m,\hat{k}}\} \tag{12}$$

where $Z^{(L)}_{b_m,k}$ denotes the embedding obtained after $L$ message passing layers.

To form the global representation for all nodes in all batches, we stack the sequences across all batches:

$$\mathcal{Z} = \big\|^{\hat{m}}_{m=1} Z_{b_m}, \tilde{X} = \big\|^{\hat{m}}_{m=1} \tilde{X}_{b_m}, \mathcal{Z}' = \tilde{X} \| \mathcal{Z} \tag{13}$$

where $\mathcal{Z}$ contains the final hop-aware embeddings for all nodes in all batches, and $\tilde{X}$ contains the original feature for all nodes after Dropout. $\mathcal{Z}'$ thus forms the global sequence input to our COMBA. Algorithm 2 in Appendix lists the pseudo-code of the cross batch aggregation process.

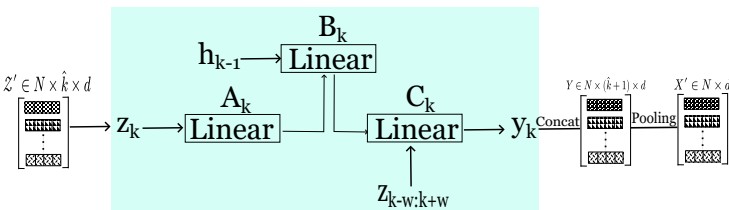

Figure 2: Illustration of the COMBA block with context gating. From left to right. The input sequence $\mathcal{Z}'$ is first processed by the S4 module to produce hop-wise representations. A context gating mechanism $\mathbf{C}$ is then applied over the local hop window $z_{k-w:k+w}$ to refine each hop embedding $y_k$. The gated outputs are concatenated with the original node features, forming a new sequence $Y$. Pooling along the hop dimension aggregates the sequence into embedding $X'$ for downstream tasks.

### 4.3 CONTEXT GATING

The COMBA framework follows the state space model (SSM) formulation in Eq. 3, but in our approach, we replace $t$ with the hop index $k$ and the input $x(t)$ with the hop-aware node embedding $z_k$. Thus, the zero-order hold discrete recurrence following Eq. 5 becomes

$$h_k = \bar{\mathbf{A}}h_{k-1} + \bar{\mathbf{B}}z_k, \quad y_k = \mathbf{C}h_k. \tag{14}$$

In the standard Mamba formulation, the output is obtained by applying a fixed projection to the hidden state. In our approach, the output projection is *hop-varing* and is computed from a local context window. For example, at hop step $k$ we may form a context window of inputs $\mathcal{Z}'_{k-1:k+1} = [z_{k-1}, z_k, z_{k+1}]$, which captures the graph context around $z_k$. More generally, as shown in Fig. 2 for a window size $w$ the input segment is $\mathcal{Z}'_{z-w:z+w} = [z_{k-w}, z_{k-k+1}, \ldots, z_{k+w}]$.

We then compute a context-dependent output matrix

$$\mathbf{C}_k = \Phi\big(\mathcal{Z}'_{k-w:k+w}\big), \tag{15}$$

where $\Phi$ is a learnable mapping. The model output at hop $k$ is obtained by applying this context-dependent projection to the current hidden state:

$$y_k = \mathbf{C}_k h_k. \tag{16}$$

Algorithm 3 in Appendix lists the pseudo-code of the context gating mechanism.

### 4.4 COMBA FRAMEWORK

Given node features $X$, a set of sampled batches $\mathcal{B}$, and their corresponding adjacency matrices $\mathcal{A}$, the COMBA framework first utilizes the CrossBatch module in ④ of Fig. 3 which aggregates information across batches to construct contextual sequences $\mathcal{Z}'$.

$$\mathcal{Z}' = \texttt{CrossBatch}(X, \mathcal{B}, \mathcal{A})$$

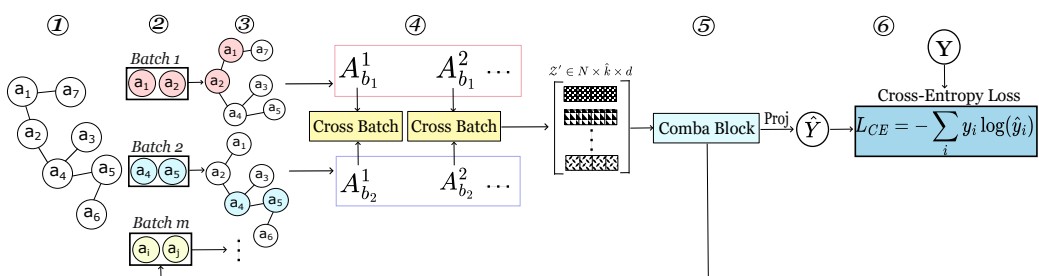

Figure 3: The proposed COMBA framework on large homogeneous graph. From left to right. Nodes of a homogeneous graph in ① are partitioned into $\hat{m}$ batches in ②. ③: for each batch, COMBA identifies the $\hat{k}$-hop neighbors of target nodes and constructs a corresponding subgraph. ④: Node embeddings will be updated across batches via cross batch aggregation as illustrated in Fig.1. The resulting sequence will pass into COMBA block with context gating ⑤ as illustrated in Fig.2. ⑥: the final predictions $\hat{Y}$ for all nodes are obtained and optimized using the cross-entropy loss

Then, these sequences are passed to the COMBA Block in ⑤ of Fig. 3 which applies context gating over different neighborhood hops to control aggregation adaptively.

$$X' = \texttt{CombaBlock}(\mathcal{Z}', \mathcal{B}, \mathcal{A})$$

Finally, model training is guided by the cross-entropy loss over the predicted labels. Algorithm 1 in Appendix lists the pseudo-code of the main COMBA framework.

### 4.5 THEORETICAL ANALYSIS

We justify that cross-batch aggregation guarantees lower error than training GNN without interactions between batches through following theorem:

**Theorem 1.** *Denote the number of batches per group as $d$, with the set of batches $\mathcal{B} = \{B_1, \ldots, B_d\}$. Let $\mathcal{BI}$ be the set of all node indices appearing in $\mathcal{B}$, and define the set of all seed node indices as*

$$\mathcal{S} = \bigcup_{i=1}^{d} s_i,$$

*where $s_i$ are the seed node indices in batch $B_i$. The complement of $\mathcal{S}$ with respect to $\mathcal{BI}$ is then $\mathcal{S}^{\mathsf{c}} = \mathcal{BI} \setminus \mathcal{S}$.*

*For each batch $B_i$ at layer $l$: - Let $A_i$ be the sampled subgraph, - Let $X_i^l$ be the node features for all nodes in $B_i$, - Let $\tilde{X}_i^l = X_i^l[s_i]$ denote features of seed nodes $s_i$, - Let $\bar{X}_i^l$ denote features of other seed nodes from different batches that appear in $B_i$, - Let $s_i^j$ be the indices of such seed nodes from batch $B_j$, and let $\kappa$ be the set of all available batches.*

*Define a gated GNN layer as a function $f(A_i, X_i^l)$. Then, the cross-batch aggregation update at layer $l+1$ is given by:*

$$\tilde{X}_i^l = f(A_i, X_i^l)[s_i] \tag{1}$$

$$\bar{X}_i^l = \big\|_{j \neq i,\, j \in \kappa} f(A_j, X_j^l)[s_i^j] \tag{2}$$

$$X_i^{l+1}[s_i] = \tilde{X}_i^l \tag{3}$$

$$X_i^{l+1}[s_i^j] = \bar{X}_i^l \tag{4}$$

*We can express the cross-batch aggregation as an approximate gated GNN layer in the following form in terms of batch $B_i$:*

$$X_i^{l+1}[n] = \mathbb{I}_i(n) \cdot f(A_i, X_i^l)[n] + (1 - \mathbb{I}_i(n)) \cdot X_i^l[n] \tag{17}$$

*where the indicator function $\mathbb{I}_i(n) \in \{0, 1\}$ is treated as a hard gate defined as:*

$$\mathbb{I}_i(n) = \begin{cases} 1, & \text{if } n \in \mathcal{S}_i \\ 0, & \text{if } n \in \mathcal{S}_i^{\mathsf{c}} \end{cases} \tag{18}$$

*Define the ideal aggregation update when trained over full batched graph for the set of batch $\mathcal{B}$:*

$$\bar{X}_i^{l+1}[n] = f(A_i, X_i^l)[n] \tag{19}$$

*Define the aggregation update without cross-batch update for the set of batch $\mathcal{B}$ as $\hat{X}_i^{l+1}$. Define the approximation error between the cross-batch aggregation update and the ideal aggregation update when trained over full graph as:*

$$\mathcal{E}^{l+1}(X_i^{l+1}, \bar{X}_i^{l+1}) = \frac{1}{d} \sum_{i=1}^d \frac{1}{|B_i|} \sum_{j \in B_i} \|X_i^{l+1} - \bar{X}_i^{l+1}\|_2^2 \tag{20}$$

*We can show that $\mathcal{E}^{l+1}(X_i^{l+1}, \bar{X}_i^{l+1}) \leq \mathcal{E}^{l+1}(\hat{X}_i^{l+1}, \bar{X}_i^{l+1})$.*

The proof of Theorem 1 is presented in Appendix due to space limit.

## 5 Experiments

### 5.1 Benchmark Datasets

Six real-world homogeneous graphs are used as our benchmark datasets. We used four heterophilic datasets from the work (Platonov et al., 2023) and two large datasets from Open Graph Benchmark (Hu et al., 2020). Additional details about the datasets are shown in Appendix.

### 5.2 Baselines

We compare our COMBA with (1) GNN, e.g., GCN (Kipf & Welling, 2017) and GatedGCN (Bresson & Laurent, 2017), (2) a transformer architecture: NAGphormer (Chen et al., 2022), (3) a scalable GNN: ClusterGCN (Chiang et al., 2019), and two recent variants of Graph Mamba: Graph Mamba-1 Behrouz & Hashemi (2024) and Graph Mamba-2 Wang et al. (2024). Additional details of the baseline models are in Appendix.

### 5.3 Results and Analysis

**Baseline Comparison** Table 1 summarizes the performance of various models on six homogeneous graph datasets. COMBA consistently achieves the highest accuracy across all datasets, outperforming baseline models. Under the same message passing scheme, COMBA shows statistically significant improvements on five datasets, including Roman-empire, Ogbn-arxiv, Ogbn-product, Minesweeper, and Tolokers. On the Amazon-ratings dataset, COMBA achieves the highest accuracy while Nagphormer performs comparably but requires more memory, highlighting COMBA's robustness across various graph structures.

Notably, COMBA consistently outperforms Graph Mamba-1 across five benchmark datasets. While Graph Mamba-1 constructs multiple graphs for each node, which can introduce redundancy and increase computational complexity, COMBA efficiently aggregates hop-aware node embeddings within a single unified graph. This design allows COMBA to capture structural information effectively, resulting in higher accuracy and more reliable performance compared to Graph Mamba-1.

**Abalation Study** Table 2 presents the ablation results of COMBA, showing that each component plays an important role in its overall performance. When the cross-batch aggregation is removed, the model consistently underperforms, underscoring the necessity of aggregating information across batches to capture richer global structural patterns. Likewise, removing the context gating mechanism leads to a further drop in accuracy, which demonstrates its effectiveness in filtering and emphasizing relevant hop-level information. These results confirm that both cross-batch aggregation and context gating are integral to COMBA's strong performance. In particular, Amazon-ratings and Tolokers exhibit substantial gains when both the cross batch aggregation and context gating mechanism are applied, underscoring their effectiveness in settings with dense connectivity and feature noise.

Table 1: Performance comparisons between baselines and our proposed method across five homogeneous datasets. Over 5 different initialization status, accuracies are reported for Roman-empire, Amazon-ratings, Ogbn-arxiv, and Ogbn-products and roc auc scores are reported for Minesweeper and Tolokers. Superscript * indicates that COMBA is statistically significantly better than this method at 95% confidence level using the performance metrics.

| Model | Roman-empire Accuracy | Amazon-ratings Accuracy | Ogbn-arxiv Accuracy | Ogbn-product Accuracy | Minesweeper ROC AUC | Tolokers ROC AUC |
|---|---|---|---|---|---|---|
| GCN | $0.795^*_{\pm 0.0088}$ | $0.462_{\pm 0.0080}$ | $0.680^*_{\pm 0.0044}$ | OOM | $0.884^*_{\pm 0.0046}$ | $0.838^*_{\pm 0.0034}$ |
| Gated-GCN | $0.833^*_{\pm 0.0079}$ | $0.481^*_{\pm 0.0071}$ | $0.701^*_{\pm 0.0065}$ | OOM | $0.905^*_{\pm 0.0016}$ | $0.832^*_{\pm 0.0090}$ |
| Nagphormer | $0.800^*_{\pm 0.0043}$ | $0.504_{\pm 0.0070}$ | $0.696^*_{\pm 0.0023}$ | OOM | $0.903^*_{\pm 0.0006}$ | $0.834^*_{\pm 0.0045}$ |
| Cluster-GCN | $0.815^*_{\pm 0.0042}$ | $0.476^*_{\pm 0.0047}$ | $0.678^*_{\pm 0.0076}$ | $0.717_{\pm 0.0053}$ | $0.886^*_{\pm 0.0024}$ | $0.758^*_{\pm 0.0056}$ |
| Graph Mamba-1 | $0.677^*_{\pm 0.0009}$ | $0.415^*_{\pm 0.0016}$ | $0.604^*_{\pm 0.0020}$ | OOM | $0.806^*_{\pm 0.0059}$ | $0.734^*_{\pm 0.0113}$ |
| Graph Mamba-2 | $0.869^*_{\pm 0.0092}$ | $0.490^*_{\pm 0.0036}$ | $0.686^*_{\pm 0.0089}$ | OOM | $0.927^*_{\pm 0.0021}$ | $0.803^*_{\pm 0.0199}$ |
| COMBA | $\mathbf{0.895}_{\pm 0.0038}$ | $\mathbf{0.507}_{\pm 0.0025}$ | $\mathbf{0.716}_{\pm 0.0037}$ | $\mathbf{0.735}_{\pm 0.0100}$ | $\mathbf{0.942}_{\pm 0.0039}$ | $\mathbf{0.845}_{\pm 0.0009}$ |

Table 2: Ablation study results *w.r.t.* cross batch aggregation and context gating mechanism

| Model | Roman-empire Accuracy | Amazon-ratings Accuracy | Ogbn-arxiv Accuracy | Minesweeper ROC AUC | Tolokers ROC AUC |
|---|---|---|---|---|---|
| COMBA | $\mathbf{0.895}_{\pm 0.0038}$ | $\mathbf{0.507}_{\pm 0.0025}$ | $\mathbf{0.716}_{\pm 0.0037}$ | $\mathbf{0.942}_{\pm 0.0039}$ | $\mathbf{0.845}_{\pm 0.0009}$ |
| w/o cross batch | $0.881_{\pm 0.0040}$ | $0.492_{\pm 0.0062}$ | $0.703_{\pm 0.0009}$ | $0.940_{\pm 0.0051}$ | $0.832_{\pm 0.0022}$ |
| w/o context gating | $0.878_{\pm 0.0005}$ | $0.472_{\pm 0.0007}$ | $0.705_{\pm 0.0014}$ | $0.939_{\pm 0.0050}$ | $0.805_{\pm 0.0039}$ |

**Complexity Analysis** We evaluate the wall-clock runtime across six benchmark datasets to assess the scalability of our approach. As illustrated in Figure 4, the average training time per epoch consistently increases with the total number of nodes and edges. Importantly, this relationship follows an approximately linear trend on a log-log scale, indicating that our method maintains efficient scalability even on large homogeneous graphs. This trend suggests that our proposed framework avoids the neighborhood explosion problem often encountered in GNN. Moreover, this observed linear behavior demonstrates that our cross batch aggregation and context-gating mechanism do not introduce computational overhead. This

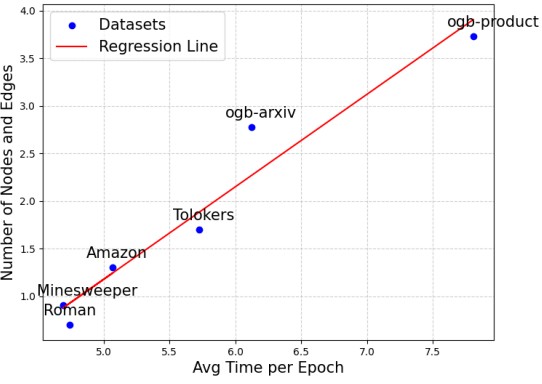

Figure 4: Log average runtime per epoch ($y$-axis) using fixed batch sizes *vs.* the sum of number of nodes and number of edges in log scale ($x$-axis).

scalability is especially important for real-world applications, where graphs with millions of nodes and edges are common, underscoring the utility of our approach in large-scale graph scenarios.

## 6 CONCLUSION

This paper introduces COMBA, a state space model–based framework for large graph learning. Unlike prior methods that inefficiently obtain subgraphs for each node, COMBA uses multi-hop neighbors of each batch to construct a subgraph for scalable GNN learning. In contrast to approaches that only consider the current step during gating, our context gating mechanism captures richer graph contexts from a window of neighborhoods. To mitigate the bias introduced by batch sampling and capture global structural information, we employ cross-batch aggregation that promotes information switch between batches with theoretical guarantees for error reduction. Together, these innovations enable COMBA to scale to large graphs while achieving robustness and superior performance. Extensive experiments on benchmark datasets confirm that COMBA consistently outperforms baseline methods, highlighting its effectiveness in capturing long-range dependencies with improved accuracy, efficiency, and robustness.

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

# A APPENDIX

## A.1 THEOREM PROOF

**Theorem.** *Denote the number of batches per group as $d$, with the set of batches $\mathcal{B} = \{B_1, \ldots, B_d\}$. Let $\mathcal{BI}$ be the set of all node indices appearing in $\mathcal{B}$, and define the set of all seed node indices as*

$$\mathcal{S} = \bigcup_{i=1}^{d} s_i,$$

*where $s_i$ are the seed node indices in batch $B_i$. The complement of $\mathcal{S}$ with respect to $\mathcal{BI}$ is*

$$\mathcal{S}^{\mathrm{c}} = \mathcal{BI} \setminus \mathcal{S}.$$

*For each batch $B_i$ at layer $l$: - Let $A_i$ be the sampled subgraph, - Let $X_i^l$ be the node features for all nodes in $B_i$, - Let $\tilde{X}_i^l = X_i^l[s_i]$ denote features of seed nodes $s_i$, - Let $\bar{X}_i^l$ denote features of other seed nodes from different batches that appear in $B_i$, - Let $s_i^j$ be the indices of such seed nodes from batch $B_j$, and let $\kappa$ be the set of all available batches.*

*Define a gated GNN layer as a function $f(A_i, X_i^l)$.*

*Then, the cross-batch aggregation update at layer $l + 1$ is given by:*

$$\tilde{X}_i^l = f(A_i, X_i^l)[s_i] \tag{1}$$

$$\bar{X}_i^l = \big\|_{j \neq i,\, j \in \kappa} f(A_j, X_j^l)[s_i^j] \tag{2}$$

$$X_i^{l+1}[s_i] = \tilde{X}_i^l \tag{3}$$

$$X_i^{l+1}[s_i^j] = \bar{X}_i^l \tag{4}$$

*We can express the cross-batch aggregation as an approximate gated GNN layer in the following form in terms of batch $B_i$:*

$$X_i^{l+1}[n] = \mathbb{I}_i(n) \cdot f(A_i, X_i^l)[n] + (1 - \mathbb{I}_i(n)) \cdot X_i^l[n] \tag{21}$$

*where the indicator function $\mathbb{I}_i(n) \in \{0, 1\}$ is treated as a hard gate defined as:*

$$\mathbb{I}_i(n) = \begin{cases} 1, & \text{if } n \in \mathcal{S}_i \\ 0, & \text{if } n \in \mathcal{S}_i^{\mathsf{c}} \end{cases} \tag{22}$$

*Define the ideal aggregation update when trained over full batched graph for the set of batch $\mathcal{B}$:*

$$\bar{X}_i^{l+1}[n] = f(A_i, X_i^l)[n] \tag{23}$$

*Define the aggregation update without cross-batch update for the set of batch $\mathcal{B}$ as $\hat{X}_i^{l+1}$. Define the approximation error between the cross-batch aggregation update and the ideal aggregation update when trained over full graph as:*

$$\mathcal{E}^{l+1}(X_i^{l+1}, \bar{X}_i^{l+1}) = \frac{1}{d} \sum_{i=1}^{d} \frac{1}{|B_i|} \sum_{j \in B_i} \|X_i^{l+1} - \bar{X}_i^{l+1}\|_2^2 \tag{24}$$

*We can show that $\mathcal{E}^{l+1}(X_i^{l+1}, \bar{X}_i^{l+1}) \leq \mathcal{E}^{l+1}(\hat{X}_i^{l+1}, \bar{X}_i^{l+1})$.*

**Proof 1.** *Fix a batch $i$ and a seed node $n \in B_i$. Define the per-node squared errors relative to the ideal update:*

$$E_X(n) = \|X_i^{l+1}[n] - \bar{X}_i^{l+1}[n]\|_2^2, \qquad E_{\hat{X}}(n) = \|\hat{X}_i^{l+1}[n] - \bar{X}_i^{l+1}[n]\|_2^2.$$

**Case 1.** $n \in s_i$ *(local seed nodes). For local seed nodes, the cross-batch update and the ideal update coincide:*

$$X_i^{l+1}[n] = \bar{X}_i^{l+1}[n].$$

*Hence $E_X(n) = 0 \leq E_{\hat{X}}(n)$.*

**Case 2.** $n \in s_i^{\mathsf{c}}$ *(cross-batch nodes). Let $a(n) = \bar{X}_i^{l+1}[n] = f(A_i, X_i^l)[n]$ be the ideal update, $b(n) = X_i^{l+1}[n]$ the cross-batch update, and $c(n) = \hat{X}_i^{l+1}[n]$ the no cross-batch value.*

*The squared errors relative to the ideal update are*

$$E_X(n) = \|b(n) - a(n)\|_2^2, \qquad E_{\hat{X}}(n) = \|c(n) - a(n)\|_2^2.$$

*Substituting $b(n)$ gives*

$$b(n) - a(n) = (\mathbb{I}_i(n) - 1)a(n) + (1 - \mathbb{I}_i(n))X_i^l[n] = (1 - \mathbb{I}_i(n))\big(X_i^l[n] - a(n)\big),$$

*so that*

$$E_X(n) = (1 - \mathbb{I}_i^{cross}(n)) \|X_i^l[n] - a(n)\|_2^2.$$

*Similarly, for the no-cross-batch update,*

$$E_{\hat{X}}(n) = (1 - \mathbb{I}_i^{no-cross}(n)) \|X_i^l[n] - a(n)\|_2^2.$$

Since $\mathbb{I}_i^{cross}(n) \geq \mathbb{I}_i^{no\text{-}cross}(n), \quad \forall n \in B_i$, it follows that

$$E_X(n) = (1 - \mathbb{I}_i^{cross}(n)) \|X_i^l[n] - a(n)\|_2^2 \leq (1 - \mathbb{I}_i^{no-cross}(n)) \|X_i^l[n] - a(n)\|_2^2 = E_{\hat{X}}(n).$$

**Conclusion.** For every node $n$, we have $E_X(n) \leq E_{\hat{X}}(n)$. Averaging over all nodes and batches,

$$\mathcal{E}^{l+1}(X^{l+1}, \bar{X}^{l+1}) = \frac{1}{d} \sum_{i=1}^{d} \frac{1}{|B_i|} \sum_{n \in B_i} E_X(n) \leq \frac{1}{d} \sum_{i=1}^{d} \frac{1}{|B_i|} \sum_{n \in B_i} E_{\hat{X}}(n) = \mathcal{E}^{l+1}(\hat{X}^{l+1}, \bar{X}^{l+1}).$$

*Thus the inequality holds.*

### A.2 DETAILED IMPLEMENTATIONS

We provide the pseudo code in Algorithm 1 to show the full pipeline of COMBA

---

**Algorithm 1:** Comba

**Input** : Node features $\{X\}$,
  Batch set $\mathcal{B} = \{b_1, \ldots, b_{\hat{m}}\}$,
  All batches' sequences of matrices $\mathcal{A} = \{A_{b_1}, \ldots, A_{b_{\hat{m}}}\}$
**Output:** $\hat{Y}$ for downstream prediction tasks

1 $\mathcal{Z}' \leftarrow \texttt{CrossBatch}(X, \mathcal{B}, \mathcal{A})$
2 $X' \leftarrow \texttt{CombaBlock}(\mathcal{Z}', \mathcal{B}, \mathcal{A})$
3 $\hat{Y} \leftarrow \texttt{Projection}(X')$
4 $\mathcal{L} \leftarrow - \sum_i y_i \log(\hat{y}_i)$ ;  // Calculate cross-entropy loss
5 **return** $\hat{Y}$

---

**Algorithm 2:** Cross batch aggregation

**Input** : Node features $\{X\}$,
  Batch set $\mathcal{B} = \{b_1, \ldots, b_{\hat{m}}\}$,
  All batches' sequences of matrices $\mathcal{A} = \{A_{b_1}, \ldots, A_{b_{\hat{m}}}\}$
**Output:** Sequence $\mathcal{Z}' \in N \times (\hat{k} + 1) \times d$ for nodes in all batches

1 **for** *each batch $b_m$ in $\{b_1, \ldots, b_{\hat{m}}\}$* **do**
2   $\tilde{X}_{b_m} \leftarrow \text{Dropout}(X_{b_m}, p)$
3   $Z_{b_m}^{(0)} \leftarrow \texttt{Projection}(\tilde{X}_{b_m})$
4   **for** *each $A_{b_m}^k$ in $A_{b_m}$* **do**
      /* Cross-batch update for each node n in batch $b_m$                                */
5     $Z_{b_{1:\hat{m}},k}^{(0)}(n) \leftarrow GNN(Z_{b_m}^{(0)}, A_{b_m}^k)(n), \quad \forall n \in b_m \cap b_{1:\hat{m}}$
6     **for** $l = 0, \ldots, L-1$ **do**
7       $Z_{b_m,k}^{(l+1)} \leftarrow GNN(Z_{b_m,k}^{(0)}, A_{b_m}^k)$
8       $Z_{b_{1:M},k}^{(0)}(n) \leftarrow Z_{b_m,k}^{(l+1)}(n), \quad \forall n \in b_m \cap b_{1:\hat{m}}$
9     **end**
10   **end**
11   $Z_{b_m} \leftarrow (Z_{b_m,1}^{(L)}, Z_{b_m,2}^{(L)}, \ldots, Z_{b_m,\hat{k}}^{(L)})$ ;  // Obtain sequence for batch $b_m$
12 **end**
13 $\mathcal{Z} \leftarrow \big\|_{m=1}^{\hat{m}} Z_{b_m}$ ;  // Obtain sequences for all batches
14 $\tilde{X} \leftarrow \big\|_{m=1}^{\hat{m}} \tilde{X}_{b_m}$
15 $\mathcal{Z}' \leftarrow \tilde{X} \| \mathcal{Z}$
16 **return** $\mathcal{Z}'$

---

---

**Algorithm 3:** COMBA Block for Node Embeddings

---

**Input** : Node embeddings sequences $\mathcal{Z} \in \mathbb{R}^{N \times \hat{k} \times d}$,
  Batch set $\{b_1, \ldots, b_{\hat{m}}\}$,
  All batches' sequences of matrices $\{A_{b_1}, \ldots, A_{b_{\hat{m}}}\}$
**Output:** Updated node embeddings $X' \in \mathbb{R}^{N \times d}$

1 **for** *each batch $b_m$ **in** $\{b_1, \ldots, b_{\hat{m}}\}$ and $l = 0, \ldots, L$* **do**
2  $\quad \mathcal{Z}' \leftarrow \texttt{Layernorm}(\mathcal{Z}^{(l)}_{b_m})$
3  $\quad \mathbf{A} \leftarrow \texttt{Linear}_A(\mathcal{Z}')$
4  $\quad \mathbf{B} \leftarrow \texttt{Linear}_B(\mathcal{Z}')$
5  $\quad \mathbf{C} \leftarrow \texttt{Linear}_C(\mathcal{Z}'_{k-w:k+w})(Eq.\ 15)$ ; $\quad$ // Compute context-dependent output matrix
6  $\quad \Delta \leftarrow \texttt{softplus}(\texttt{Linear}_\Delta(\mathcal{Z}_{b_m}))$
7  $\quad \bar{\mathbf{A}} \leftarrow \texttt{discretize}(\Delta, A)(Eq.\ 11)$
8  $\quad \bar{\mathbf{B}} \leftarrow \texttt{discretize}(\Delta, A, B)(Eq.\ 12)$
  $\quad$ /* Use context gating mechanism to produce output at each hop $k$ $\qquad\qquad$ */
9  $\quad$ **for** $n = 1$ **to** $N$ **do**
10  $\quad\quad (x_1, \ldots, x_{\hat{k}}) \leftarrow \mathcal{Z}'_{n,:,:}$
11  $\quad\quad h_0 \leftarrow 0$
12  $\quad\quad$ **for** $k = 1$ **to** $\hat{k}$ **do**
13  $\quad\quad\quad h_k \leftarrow \bar{\mathbf{A}}_k h_{k-1} + \bar{\mathbf{B}}_k x_k$
14  $\quad\quad\quad y_k \leftarrow \mathbf{C}_k h_k$
15  $\quad\quad$ **end**
16  $\quad$ **end**
17  $\quad Y : (|b_m|, \hat{k}, d) \leftarrow \|_{k=1}^{\hat{k}} y_k$ ; $\quad$ // Output sequence for all nodes in batch $b_m$
18  $\quad Y' \leftarrow \texttt{LayerNorm}\big(Y + \mathcal{Z}_{b_m}\big)$
19  $\quad X'_{b_m} \leftarrow \frac{1}{\hat{k}} \sum_{k=1}^{\hat{k}} Y'_{:,k,:}$ ; $\quad$ // Pooling across hop dimension
20  $\quad$ **for** *each $A^k_{b_m}$ **in** $A_{b_m}$* **do**
21  $\quad\quad Y^k_{b_m} \leftarrow GNN(X'_{b_m}, A^k_{b_m})$
22  $\quad$ **end**
23  $\quad Y_{b_m} \leftarrow (Y^1_{b_m}, Y^2_{b_m}, \ldots, Y^{\hat{k}}_{b_m})$ ; $\quad$ // Rebuild sequence
24  $\quad \mathcal{Z}^{(l+1)}_{b_m} \leftarrow X'_{b_m} \| Y_{b_m}$ ; $\quad$ // Concatenate node representations $X'_{b_m}$ with sequence $Y_{b_m}$
25 **end**
26 $X' \leftarrow \|_{m=1}^{\hat{m}} X'_{b_m}$ ; $\quad$ // Concatenate node representations for nodes in all batches
27 **return** $X'$

---

## A.3 DETAILS OF DATASETS

Table 3: Dataset Statistics

|  | Roman-empire | Amazon-ratings | Minesweeper | Tolokers | Ogbn-arxiv | Ogbn-products |
|---|---|---|---|---|---|---|
| # Nodes | 22,662 | 24,492 | 10,000 | 11,758 | 169,343 | 2,449,029 |
| # Edges | 32,927 | 93,050 | 39,402 | 51,900 | 1,166,243 | 61,859,140 |
| # Features | 300 | 300 | 7 | 10 | 128 | 100 |
| # Classes | 18 | 5 | 2 | 2 | 40 | 47 |

**Roman-empire(Platonov et al., 2023):** This dataset is based on the Roman Empire article from Wikipedia. Each node in the graph represents one word in the text, and edges exist if two words are connected. For semi-supervised learning, the nodes are split into training, validation, and test sets with 11,331 (50%), 5,665 (25%), and 5,666 (25%) nodes, respectively.

**Amazon-ratings(Platonov et al., 2023):** This dataset is based on the Amazon product co-purchasing network. Each node in the graph represents products, and edges exist if products are bought together. For semi-supervised learning, the nodes are split into training, validation, and test sets with 12,246 (50%), 6,123 (25%), and 6,123 (25%) nodes, respectively.

**Minesweeper(Platonov et al., 2023):** This dataset inspired by the Minesweeper game, a synthetic dataset. Each node in the graph represents one cell in the grid, and edges exist for neighboring cells. For semi-supervised learning, the nodes are split into training, validation, and test sets with 5,000 (50%), 2,500 (25%), and 2,500 (25%) nodes, respectively.

**Tolokers(Platonov et al., 2023):** This dataset is based on data from the Toloka crowdsourcing platform. Each node in the graph represents tolokers (workers), and edges exist if two tolokers work on the same task. For semi-supervised learning, the nodes are split into training, validation, and test sets with 5,879 (50%), 2,939 (25%), and 2,940 (25%) nodes, respectively.

**Ogbn-arxiv(Hu et al., 2020):** This dataset is from Open Graph Benchmark, representing the citation network for computer science papers. Each node in the graph represents a paper, and edges exist if one paper cites another one. For semi-supervised learning, the nodes are split into training, validation, and test sets with 90,941 (53.7%), 29,799 (17.6%), and 48,603 (28.7%) nodes, respectively.

**Ogbn-products(Hu et al., 2020):** This dataset is also from Open Graph Benchmark, representing an Amazon product co-purchasing network. Each node in the graph represents products sold in Amazon, and edges exist if two products are purchased together. For semi-supervised learning, the nodes are split into training, validation, and test sets with 196,615 (8.0%), 39,323 (1.6%), and 2,213,091 (90.4%) nodes, respectively.

## A.4 DETAILS OF BASELINE MODELS

We compare our COMBA with some state-of-art baselines.

**GCN** (Kipf & Welling, 2017) is a homogeneous graph neural network. It learns node representations by aggregating and transforming features from each node's neighbors.

**Gated-GCN** (Bresson & Laurent, 2017) extends standard GCN by incorporating gating mechanisms to control the flow of information from neighboring nodes, allowing more flexible and selective feature aggregation. Our COMBA employs the Gated-GCN model as default for GNN.

**Nagphormer** (Chen et al., 2022) is a graph transformer model. It introduces the Hop2Token module, which aggregates neighborhood features from multiple hops into distinct token representations.

**Cluster-GCN** (Chiang et al., 2019) partitions the graph into clusters and performing mini-batch training within these clusters. By reducing the neighborhood size per batch, it preserves graph structure while lowering memory and computational costs.

**Graph Mamba-1** (Behrouz & Hashemi, 2024) treats each node's induced subgraphs as tokens and constructs input sequences by processing these subgraphs with MPNNs. Since it is not reproducible due to the high time complexity of the full model, we adopt a simplified version for efficiency.

**Graph Mamba-2** (Wang et al., 2024) leverages SSM to efficiently capture long-range dependencies in graphs. By permutation and node prioritization techniques, it achieves strong predictive performance with reduced computational and memory costs.

## A.5 IMPLEMENTATION DETAILS

We conduct a grid search over a selected range of hyperparameters, including hidden dimension: [64,128], number of layers: [2,3], feature dropout rate: [0,0.3,0.5], hop length: [2,3,5,10] and batch size: [5,10,50,100]. Adam (Kingma & Ba, 2014) is used as the optimizer. The learning rate, weight decay, and number of training epochs are fixed, with early stopping applied. For each method, we report the average accuracy over five different random seeds. All experiments are performed on desktop workstations equipped with NVIDIA RTX A6000 Ada Generation GPUs.

## A.6 THE USE OF LLMS

LLMs are used to generate initial code skeletons in the research process. These drafts are later refined, debugged, and adapted to the specific requirements of our COMBA framework.

