# OpenReview forum: "COMBA: Cross Batch Aggregation for Learning Large Graphs with Context Gating State Space Models"
_ICLR.cc/2026/Conference — ICLR 2026 Conference Withdrawn Submission_

### Official Review · Reviewer_CaJ8 · 2025-10-29

**Soundness:** 2
**Presentation:** 2
**Contribution:** 2
**Rating:** 2
**Confidence:** 5

**Summary:**

This paper proposes COMBA, a framework that integrates SSMs with GNNs for large-scale graph learning. The key contributions are twofold: Cross-Batch Aggregation and Graph Context Gating.

**Strengths:**

1. Scaling SSMs to large graphs is a relevant and timely research direction, given the success of SSMs in sequence modeling.

2. Comprehensive Evaluation on six datasets of varying sizes (from ~10K to ~2.4M nodes) with consistent improvements over baselines.

3. The batch-based approach with cross-batch information sharing is intuitive and addresses real scalability concerns.

**Weaknesses:**

1. The motivation for introducing SSMs in graph learning is not clearly articulated. The original Mamba paper proposed SSMs as a means to achieve linear complexity while maintaining performance in sequence modeling. However, this work directly applies SSMs to graphs without explaining the expected benefits compared to Graph Transformers. I suggest the authors explicitly clarify why SSMs are adopted in the graph domain and what advantages (e.g., efficiency, stability, scalability) they bring, ideally supported by a formal efficiency analysis.

2. Another concern is about novelty. The only novel component in this paper appears to be the proposed Cross-Batch Aggregation. However, this idea is quite incremental. CBA essentially performs graph partitioning or clustering prior to training and aggregates information via inter-cluster edges—an approach widely explored in scalable GNN frameworks such as Cluster-GCN[1] and GAS[2]. Other parts of the model, such as context gating, are standard operations in sequence modeling and do not add substantial novelty.

3. Theorem 1 is conceptually trivial—it restates the intuitive fact that overlapping batches can reduce approximation error compared to disjoint ones. This is well known in large-scale graph learning, where increasing the number of nodes reduces bias. Furthermore, the theorem defines a full-batch aggregation (line 378), which is inconsistent with the mini-batch setup used in the actual experiments. The theoretical section therefore adds limited value.

4. The current formulation seems to focus exclusively on homogeneous graphs. It remains unclear whether the proposed approach can generalize to heterogeneous graphs. This limits the method’s applicability in real-world settings.

5. Line 209 implies that separate GNNs may be trained for each batch. If this interpretation is correct, the computational cost could become prohibitive. The authors should clarify the training scheme and confirm whether model parameters are shared across batches.

6. Some important hyperparameters are not well specified. For instance, how is the parameter k chosen in the context gating module? Is it tuned per dataset, or fixed globally?

7. Although the paper claims to target large-scale GNN learning, it omits several key scalable GNN baselines such as GAS, LazyGNN[3], and other recent efficient frameworks that demonstrate superior scalability and performance. Without these comparisons, it is difficult to assess the real contribution of COMBA.

8. On certain datasets, the reported results are noticeably lower than those of strong baselines (e.g., Cluster-GCN, GraphSAINT[4], and other scalable GNNs), but these baselines are not reported in the paper. This omission weakens the experimental credibility and leaves open questions about relative performance.

9. Despite emphasizing scalability, the paper does not provide a clear comparison of runtime, memory usage, or computational complexity with other scalable GNNs. Efficiency—being one of the main motivations—should be quantitatively demonstrated.

10. The datasets used are relatively small. To validate scalability claims, I recommend evaluating on large-scale benchmarks such as OGBN-papers100M or MAG240M.

11. The experiments focus only on node classification. It would strengthen the paper to include results on other important graph learning tasks such as link prediction and graph classification.

[1] Cluster-GCN: An Efficient Algorithm for Training Deep and Large Graph Convolutional Networks

[2] GNNAutoScale: Scalable and Expressive Graph Neural Networks via Historical Embeddings

[3] Lazygnn: Large-scale graph neural networks via lazy propagation

[4] Graphsaint: Graph sampling based inductive learning method

**Questions:**

Please refer to the weakness.

---

### Official Review · Reviewer_t7Fe · 2025-10-30

**Soundness:** 2
**Presentation:** 3
**Contribution:** 2
**Rating:** 2
**Confidence:** 3

**Summary:**

This paper proposes COMBA, a novel framework designed to apply State Space Models (SSMs), such as Mamba, to the task of learning on large-scale graphs. The core motivation is to leverage the linear-time complexity of SSMs while adapting them to the non-sequential, neighborhood-centric structure of graph data.

**Strengths:**

1.  **Novel Motivation and Problem Formulation:** The paper does an excellent job of identifying a key challenge in applying SSMs to graphs: the fundamental mismatch between the 1D sequential dependency of SSMs and the multi-hop neighborhood dependency of graphs.
2.  **Interesting Core Mechanisms:** Both of the paper's core contributions are well-motivated.
    * "Cross Batch Aggregation" is a practical approach to scaling GNNs, and the theoretical justification (Theorem 1) for its error reduction is a good addition.
    * "Graph Context Gating" is an intelligent modification of the Mamba architecture, which elegantly repurposes the sequence-modeling capability of SSMs to handle multi-hop graph contexts.

**Weaknesses:**

1.  **Lacking Efficiency Baselines:** The paper's main motivation is to "tackle large graph learning" and scale SSMs. However, the complexity analysis (Section 5.3, Figure 4) is limited *only* to the proposed COMBA model. There is **zero** comparison against any baseline in terms of wall-clock training time, inference time, or memory usage.
    * The authors state that baselines like Nagphormer and Graph Mamba-1/2 ran "OOM" (Out of Memory) on Ogbn-product, which supports their memory-efficiency claim.
    * However, **ClusterGCN (a key scalability baseline) did *not* OOM**, yet it is *excluded* from all efficiency/complexity comparisons. This is a major omission. The scalability claim is therefore entirely unsubstantiated.

2.  **Superficial Ablation Study:** The ablation study (Table 2) is too simple. It only demonstrates that removing the two core components degrades performance. This is a weak form of ablation. A more rigorous study would have included:
    * A comparison of "Graph Context Gating" against simpler gating alternatives (e.g., a standard fixed-projection Mamba, or a gate that only depends on the current hop $Z'_k$).
    * An analysis of the GNN backbone (the paper uses Gated-GCN) to show if the COMBA framework is robust to different GNN choices (e.g., GCN, GAT).

3.  **No Hyperparameter Sensitivity Analysis:** This is a crucial flaw. The model introduces at least two key new hyperparameters: the hop length `k` (how many hops to unroll into a sequence) and the context window size `w` (for the gating mechanism). The paper provides **no analysis** of how model performance changes with different values of `k` and `w`. It is impossible to know if the model is robust or if its performance is an artifact of extensive, un-falsifiable tuning.

**Questions:**

1.  Why were no efficiency baselines (training time, memory usage) included in the complexity analysis? A direct comparison against ClusterGCN, which did not run OOM on Ogbn-product, is essential to validate the scalability claims.
2.  Can the authors provide a hyperparameter sensitivity analysis for the hop length `k` and the context window size `w`?

---

### Official Review · Reviewer_Et1w · 2025-10-30

**Soundness:** 2
**Presentation:** 3
**Contribution:** 1
**Rating:** 2
**Confidence:** 4

**Summary:**

The paper proposes COMBA, a framework for learning large graphs that combines  3 key components (i) hop-aware graph context, (ii) cross-batch aggregation that exchanges information between overlapping batches, and (iii) a SSM block to capture multi-hop context and long-range dependencies for large homogeneous graphs. The authors provide a theoretical claim that cross-batch aggregation reduces approximation error relative to isolated batch training. Experiments shows that COMBA consistently outpermorms the six considered baselines on six small to medium size datasets.

**Strengths:**

- The paper addresses the important problem of learning on large graphs, which still is an open-challenge in graph presentation learning.
- The paper is clear and well written, despite several typos are present in the text.
- Figures and pseudocodes help the reader understand the pipeline and the overall architecture

**Weaknesses:**

Technical aspects lack clarity or are insufficiently detailed:
- The cost of computing multi-hop adjacencies $A_{b_n}^k$ can be expensive on large graphs. Authors should clarify whether $A_{b_n}^k$ is precomputed at preprocessing time and eventually provide runtimes.
- line 213, it is unclear whether the GNN (i.e., its weights $W_{b_n}$) is shared across batches and across hops within a layer.
- line 221, the window considers values from $k-w$ to $k+w$, thus its size should be $2w+1$. Authors should clarify if half window is used.
- Authors should clarify how the SSM in Eq. 3, which adopts a Single-Input, Single-Output (SISO) configuration, can deal with the multidimensional input of node states.
- Authors should clarify if batches are fixed across epochs, i.e., they always contains the same sets of nodes, or if nodes are dynamically sampled at each epoch. In the latter case, authors should clarify additional computational costs of this operation.
- The appendix only briefly mentions hyperparameter tuning, making it unclear whether all models are tuned under the same range of hyperparameters. Moreover, it is unclear if the authors employed the same experimental setting of Platonov et al., 2023, and Hu et al., 2020.

Insufficient comparison:
- Recent advances in SSMs for graph processing are not discussed or compared; the manuscript should position COMBA relative to these works [1-8].
- The multi-hop aggregation mechanism appears similar to that proposed in [5].

Limited experimental validation:
- According to the OGB website, the experimental evaluation is restricted to small- to medium-sized tasks, with only ogbn-product considered medium-scale. While it is understandable that working with graphs of the scale of ogbn-papers100M is often unfeasible for standard research labs, a broader evaluation on at least a few additional medium-sized datasets, such as Pokec and SNAP-Patents from [9], would provide a more comprehensive assessment of the method.
- The baseline coverage in the experiments is limited. The claims would be stronger if the authors included additional recent SOTA Graph SSMs (e.g., [5-8]) and SOTA models on the proposed benchmarks (e.g., [10,11]). Moreover, several of these referenced methods appear to outperform COMBA in published results.
- The authors should include ablation studies analyzing performance with respect to batch size and context window, to verify whether long-range dependencies are effectively captured and to clarify design trade-offs.
- The complexity analysis, although informative, should be empirically complemented with a comparative runtime and memory analysis against other baseline methods to better substantiate COMBA’s efficiency claims.

Minor:
- The paper contains several typos and small errors, e.g., SMM instead of SSM. We recommend carefully proofreading the manuscript.

-------

[1] [Tang et al. Modeling multivariate biosignals with graph neural networks and structured state space models. In Conference on Health, Inference, and Learning 2023](https://arxiv.org/pdf/2211.11176)

[2] [Huang et al. What can we learn from state space models for machine learning on graphs?. 2024](https://arxiv.org/pdf/2406.05815)

[3] [Zhao et al. Grassnet: State space model meets graph neural network. 2024](https://arxiv.org/pdf/2408.08583)

[4] [Li et al.  State space models on temporal graphs: A first-principles study. In NeurIPS 2024](https://openreview.net/pdf?id=UaJErAOssN)

[5] [Ding et al. Recurrent Distance Filtering for Graph Representation Learning. In ICML 2024](https://openreview.net/pdf?id=5kGfm3Pa41)

[6] [Eliasof et al. Graph Adaptive Autoregressive Moving Average Models. In ICML 2025](https://openreview.net/attachment?id=UFlyLkvyAE&name=pdf)

[7] [Ceni et al. Message-Passing State-Space Models: Improving Graph Learning with Modern Sequence Modeling. 2025](https://arxiv.org/pdf/2505.18728)

[8] [Arroyo et al. On Vanishing Gradients, Over-Smoothing, and Over-Squashing in GNNs: Bridging Recurrent and Graph Learning. 2025](https://arxiv.org/pdf/2502.10818)

[9] [Lim et al. Large Scale Learning on Non-Homophilous Graphs: New Benchmarks and Strong Simple Methods. NeurIPS 2021](https://arxiv.org/pdf/2110.14446)

[10] [Deng et al. Polynormer: Polynomial-Expressive Graph Transformer in Linear Time. In ICLR 2024](https://arxiv.org/pdf/2403.01232)

[11] [Finkelshtein et al. Cooperative Graph Neural Networks. In ICML 2024](https://arxiv.org/pdf/2310.01267)

**Questions:**

see weaknesses

---

### Official Review · Reviewer_2K7R · 2025-10-31

**Soundness:** 2
**Presentation:** 1
**Contribution:** 1
**Rating:** 0
**Confidence:** 4

**Summary:**

The paper proposes COMBA (Cross-Batch Aggregation), a framework extending [1] that aims to scale state-space models (SSMs) for large-scale graph learning. The central idea is to partition a large graph into smaller subgraphs (batches) and introduce a cross-batch aggregation mechanism that enables information exchange among batches with shared nodes. Similar to [1], COMBA applies SSM layers over the sequence of multi-hop node representations, treating the hop dimension as the sequential axis.




[1] Behrouz, Ali, and Farnoosh Hashemi. "Graph mamba: Towards learning on graphs with state space models." Proceedings of the 30th ACM SIGKDD conference on knowledge discovery and data mining. 2024.

**Strengths:**

- scaling state-space models to large-scale graphs is an important questions to study
- the proposed method shows competitive performance to previous graph state space models

**Weaknesses:**

- the major weakness of this paper is its presentation. The notation is highly ambiguous, and many key implementation details are missing. I strongly encourage the authors to substantially revise the notation and exposition, as the current version is extremely difficult to follow. For example,
    - in equation (1), $A$ is the adjancency matrix and $\text{Gen}(G, A)=A^k$ is used to represent the binary matrices that indicate whether two nodes are connected by a path of exactly length $k$. It is fairly uncommon to denote this by $A^k$, which usally means the powers of a matrix.
    - Line 203-204: it defines $A_{b_m}=\\{A\^1\_\{b\_m\}, ..., A\^\{k\}\_\{b\_m\}\\}$, but then it writes $A\^k\_\{b\_m\}=\text{Gen}(G_\{b\_m\}, A_\{b\_m\}^k)$. It does not make sense that $A^k_{b_m}$ appears on both ends. This definition is also contradictive to previous definition $\text{Gen}(G, A)=A^k$.
    - Line 217 uses $\mathcal{Z}\_\{n, k\}$ to denotes representations of node $n$ at $k$-th hop. But later at equation (9) a different notation $Z\_\{b_\{m\}, k\}(n)$ is used.
    - It is unclear how Section 4.1 and Section 4.2 are related. Section 4.2 begins with  $Z^{(0)}$ initialized from the original node features and then performs a cross-batch computation, which appears conceptually disconnected from the procedure described in Section 4.1.
    - $\hat{m}$ in equation (9) is never defined.
    - crucially, how to choose the set of batches of nodes chosen is not mentioned in the main text.
- Some of relevant literature (state space models for graphs) is absent, e.g., [2-5]


[2] Pan, Zhenyu, et al. "Hetegraph-mamba: Heterogeneous graph learning via selective state space model." arXiv preprint arXiv:2405.13915 (2024).

[3] Chen, Dexiong, Till Hendrik Schulz, and Karsten Borgwardt. "Learning long range dependencies on graphs via random walks." arXiv preprint arXiv:2406.03386 (2024).

[4] Huang, Yinan, Siqi Miao, and Pan Li. "What Can We Learn from State Space Models for Machine Learning on Graphs?." arXiv preprint arXiv:2406.05815 (2024).

[5] Zhao, Gongpei, et al. "Grassnet: State space model meets graph neural network." arXiv preprint arXiv:2408.08583 (2024).

**Questions:**

See Weaknesses.

---

### Note · Authors · 2025-11-20

**Comment:**

Thanks each reviewer's time for providing constructive comments. Our paper needs major revision, so we decide to withdraw.

**Withdrawal Confirmation:**

I have read and agree with the venue's withdrawal policy on behalf of myself and my co-authors.